# Clinical Presentation Is Dependent on Age and Calendar Year of Diagnosis in Celiac Disease: A Hungarian Cross-Sectional Study

**DOI:** 10.3390/jpm13030487

**Published:** 2023-03-08

**Authors:** Zsolt Szakács, Nelli Farkas, Enikő Nagy, Réka Bencs, Zsófia Vereczkei, Judit Bajor

**Affiliations:** 1First Department of Medicine, Medical School, University of Pécs, Ifjúság Str 13., H-7624 Pécs, Hungary; szakacs.zsolt@pte.hu; 2Institute for Bioanalysis, Medical School, University of Pécs, Szigeti Str 12, H-7624 Pécs, Hungary; nelli.farkas@aok.pte.hu; 3Department of Emergency Medicine, Medical School, University of Pécs, Ifjúság Str 13., H-7624 Pécs, Hungary; encsii66@gmail.com; 42nd Department of Internal Medicine and Nephrological Center, Medical School, University of Pécs, Pacsirta Str 1., H-7624 Pécs, Hungary; bencsreka@gmail.com; 5Institute for Translational Medicine, Medical School, University of Pécs, Szigeti Str 12, H-7624 Pécs, Hungary; vereczkei47@gmail.com; 6Department of Sport Nutrition and Hydration, Institute of Nutritional Science and Dietetics, Faculty of Health Sciences, University of Pécs, Vörösmarty Mihály Str 4., H-7621 Pécs, Hungary

**Keywords:** celiac disease, clinical phenotype, clinical presentation, calendar year, children and adults

## Abstract

International trends indicate that celiac disease (CeD) is becoming more common, while the clinical presentation of CeD tends to change. We aimed to investigate factors associated with the clinical presentation of CeD. We reviewed all CeD cases diagnosed at our tertiary center, University of Pécs (Hungary), between 1992 and 2019. We collected data of verified CeD patients on clinical presentations (classified by the Oslo Classification), the age at and calendar year of diagnosis, and sex, serology and histology at diagnosis. To assess the associations of baseline variables with clinical presentations, we applied univariate and multivariate (binary logistic regression) statistics. A total of 738 CeD patients were eligible for inclusion. In the univariate analysis, patients with classical CeD were more common in the latest calendar period (*p* < 0.001) and tended to be older (*p* = 0.056), but we failed to observe a significant association between the clinical presentation and sex, serology or histology at diagnosis. In the multivariate analysis, only age at diagnosis and calendar year were independently associated with clinical presentations (OR = 1.02, CI: 1.01–1.04 and OR = 0.93, CI: 0.89–0.98, respectively). Our findings confirmed that classical CeD is independently associated with age at diagnosis and calendar year of diagnosis of CeD, whereas other parameters were not significantly associated with clinical presentations.

## 1. Introduction

Celiac disease (CeD) is a systemic, immune-mediated disease, which develops in genetically vulnerable individuals after gluten exposure. CeD affects approximately 1% of the population worldwide [1]. The prevalence of CeD varies across continents and was reported to be high in Europe and some countries of Asia [2,3,4]. The variance in prevalence, e.g., the unexpectedly low prevalence in the far East such as in Japan and Korea, might be explained by the distribution of HLA-DQ2 and by the difference in the consumption of wheat products [5]. Parallel to the increasing wheat consumption in Japan, the prevalence of CeD is expected to rise, as well [6]. In spite of its worldwide increasing incidence, most of the patients remain undiagnosed or suffer from a significant diagnostic delay [2]. This, besides deteriorating patients’ quality of life, carries the risk of the development of many complications.

The clinical presentation of CeD has many faces including asymptomatic patients, full-blown malabsorption as well as the potentially life-threatening celiac crisis [7,8,9]. Extraintestinal signs and symptoms often develop with malabsorption or can even be present alone [10]. The list of possible extraintestinal manifestations is becoming longer; therefore, the recognition of CeD requires increased awareness [11,12].

The clinical presentations of CeD can be divided into three groups per the Oslo Classification: classical CeD (with signs and symptoms of malabsorption: diarrhea, weight loss, and growth failure), non-classical CeD (with no signs and symptoms of malabsorption), and asymptomatic CeD (with no disease-related signs and symptoms at all) [13].

The reason behind the various forms of clinical presentations of CeD has still not been clearly explained. Some patients develop severe villous atrophy with severe malabsorption; however, others with the same degree of villous damage are fully asymptomatic. Literature data on the associations across the severity and characteristics of signs and symptoms and other diagnostic parameters (e.g., age, degree of mucosal damage and titers of CeD-specific antibodies) with clinical presentations are conflicting. Theoretically, the more severe the symptoms, the more severe the villous atrophy and the higher the titers of CeD-specific antibodies [14,15,16], though this observation is not supported by all studies [17]. Sex might be associated with signs and symptoms as well; one might expect a more severe disease course and more frequent malabsorption in women [18].

Clinical presentations have been changing in recent decades, with most studies reporting an increase in the prevalence of non-classical CeD [19,20,21,22,23,24]. The change can be rooted in increasing disease awareness and disease-related knowledge as well as in better diagnostic methods leading to the increased recognition of CeD patients with non-classical or silent clinical presentations. An increase in age at diagnosis of CeD has been observed as well [19,23,25].

In this study, we investigated which baseline parameters are associated with clinical presentations of CeD.

## 2. Materials and Methods

This study is reported in accordance with the STROBE Statement. The study was conducted in accordance with the Declaration of Helsinki and approved by the Regional and Local Research Ethics Committee of University of Pécs, Pécs, Hungary (ref no. 6918).

### 2.1. Study Design and Site, Data Sources and Data Collections

This study was a single-center, retrospective cohort study. Our tertiary center at the University of Pécs (Pécs, Hungary) provides professional gastroenterological care for about 300,000 inhabitants.

### 2.2. Study Population, Eligibility and Clinical Phenotype

The flow chart of the study is presented in Figure 1. Only patients with verified CeD were eligible for inclusion, irrespective of age. The diagnosis of CeD was verified by a gastroenterologist based on a combination of clinical, serological and histopathological data per the currently valid guidelines [26,27,28,29].

The CeD patients were assessed per the Oslo Classification by clinical presentations divided into classical, non-classical, and silent CeD [13]. Diagnostic histological samples (at least 4) were taken from the distal part of the duodenum. The samples were oriented, then processed and assessed by a gastrointestinal histopathologist using the modified Marsh classification. Only if the degree of the mucosa damage was provided (partial, subtotal or total villous atrophy) did we re-classify the sample (as Marsh 3a, 3b or 3c, respectively). Commercially available ELISA kits (Orgentec Diagnostika GmbH, Mainz, Germany) for the assay of anti-tissue transglutaminase antibodies (tTG) were introduced in our clinics in 1998. A tTG level >10 U/mL was considered positive. The patients with high and low tTG titers were defined as ≥10 times or <10 times the upper limit of normal tTG, respectively. Before 1998, serological diagnosis was based on IgA anti-endomysial antibodies (EMA) detected by indirect immunofluorescence. By this method, the results were considered to be positive when a reticular pattern of immunofluorescence was observed in the muscular mucosae at a serum dilution of ≥1:5. Serum IgA levels were measured in the majority of the cases; for IgA deficiency, the diagnosis was confirmed by IgG tTG serology and intestinal histology.

Cases were excluded from the study if clinical data were insufficient for classification.

### 2.3. Data Extraction

Two earlier works, a case–control study investigating prothrombotic alterations in CeD [30] and a cohort study investigating the association between HLA haplotypes and baseline clinical variables [31], have included a fraction of this study population.

Data were retrieved (1) from prior paper-based medical files and (2) from the current medical software, eMedSolution (T-Systems Hungary Ltd., Budapest, Hungary, Version: 2023/Q1/1 (20230127151442), based on disease identifiers. The time period for the paper-based and electronic data collection started with the years 1992 and 2007, respectively, and ended with the year 2019. After identifying the patients with CeD, the data of all cases were abstracted manually by an investigator with a medical degree into a pre-defined data collection table. Then, all the data points were verified by a second investigator with a medical degree. A total of 4 persons were involved in the data collection.

The collected data included the date of birth, calendar year of the diagnosis of CeD, signs and symptoms, histology and serological results (tTG IgA and IgG, and EMA IgA and IgG) during the diagnosis of CeD. The diagnostic periods were divided into three intervals: before 1998, 1999–2007, and after 2007. The reason for this division was that the introduction of tTG measurement was in 1998 and that the introduction of the new electronic database was in 2007 in our clinic.

### 2.4. Statistical Analysis

In the analysis, we classified silent cases as non-classical CeD due to their low number (55 of 738). After completing the data collection and validation, we performed a descriptive statistical analysis. For the categorical variables, relative frequencies were calculated. For the continuous variables (age, and dates), the data distribution was checked and, accordingly, the median and range were calculated.

In the univariate comparative analysis, the groups of clinical presentations were compared with the Chi^2^-test, the Fisher’s exact test or with the Mann–Whitney test.

For the multivariate analysis, only the patients with a full dataset were eligible, being representative of the whole study population regarding age (*p* = 0.779 with the *t*-test) and sex (*p* = 0.942 with the Chi^2^-test). No imputation was performed. We used a binary logistic regression for analysis, of which the outcome variable was the clinical presentation (classical vs. non-classical) and the explanatory variables were age at diagnosis of CeD (continuous), calendar year of diagnosis of CeD (continuous), sex (binary), serology at diagnosis (tTG IgA status at diagnosis in two configurations: negative vs. positive, negative vs. low positive vs. high positive test results) and histology at diagnosis (in two configurations: Marsh 1+2 vs. Marsh 3a+b vs. Marsh 3c and Marsh 1+2 vs. Marsh 3a+b+c). All the analyses were performed in both the ‘enter’ and ‘forward selection’ modes. All the analyses were performed with the software SPSS (Version 26.0., IBM Corporation, Armonk, NY, USA).

## 3. Results

A total of 738 CeD patients were included in the analysis. The data quality is summarized in Appendix A.

The patients’ characteristics are summarized in Table 1. The mean age at the diagnosis of CeD was 22.8 years (SD: 17.1 years), and almost half (49.1%) of them were diagnosed in childhood. Approximately, one-fourth of the patients were males. Out of the 738 patients, 290 (39.3%) had classical CeD. As expected, the majority of the cases were seropositive and had villous atrophy at diagnosis. Considering the era of the diagnosis, more than two-thirds (68.6%) of the cases were diagnosed after 2007.

When comparing the baseline clinical data of the classical vs. non-classical CeD cases (Table 1), non-classical CeD became significantly more prevalent in the latest calendar period, compared to the earlier eras (*p* < 0.001 for both). Although the patients with classical CeD were older, the level of significance failed to attain the pre-specified cut-off (*p* = 0.056). The data on serology and histology at diagnosis did not significantly differ between the groups (*p* > 0.050 for all comparisons).

Out of the 738 patients, 392 (53.1%) were eligible for the multivariate analysis. Only age at diagnosis and calendar year were independently associated with clinical presentations (OR = 1.02, CI: 1.01–1.04 and OR = 0.93, CI: 0.89–0.98, respectively) (Table 2). These results were consequently observed in all the models built. This section may be divided by subheadings. It should provide a concise and precise description of the experimental results, their interpretation, as well as the experimental conclusions that can be drawn.

## 4. Discussion

Our results imply that a classical presentation was independently associated with an older age and an earlier calendar year of diagnosis. Other baseline parameters (sex, histology at diagnosis and serology at diagnosis) were not significantly associated with clinical presentations.

Our data on the association between clinical presentations and calendar year corroborate the literature data; we observed the worldwide trend of the more frequent non-classical presentation of CeD in our Hungarian cohort of patients as well [19,20,21,22,23,24,32,33,34,35,36,37,38,39]. In our clinical practice, wider access to state-of-the art diagnostic methods and better awareness of cases of non-classical CeD are assumed to be significant contributors to this trend. In the last 20 years, we have harmonized the diagnostic strategy applied in our adult and pediatric gastroenterology units with that used by specialists of other relevant disciplines (e.g., endocrinology, immunology and rheumatology) in order to comply with the recent recommendations of guidelines on active case-finding strategies [27,40].

Unexpectedly, an older age at diagnosis was significantly associated with classical presentations in our study. The literature data are rather conflicting on this matter: classical presentation still prevails only in younger children (<3 years of age) [41,42,43,44,45]. Our results might be explained by the fact that CeD patients with non-classical presentations are less frequently—and less effectively—recognized in adulthood. Unfortunately, the number of cases in patients <3 years of age did not allow us to perform a subgroup analysis to examine if the factors associated with clinical presentations of the study population still apply to this group.

Regarding the association with sex, the more severe classical clinical presentations in women were not confirmed by our study [18,21,46,47].

The most frequently examined question is the potential association of serology and histology at diagnosis with clinical presentations. One might reasonably assume that the stronger the immunological activity—indicated by the titers of CeD-specific antibodies—the more severe the symptoms. In this matter, relevant clinical data are surprisingly scarce. Pediatric research from Israel indicated that higher titers of tTG make classical presentation more likely [34]. In a Hungarian cohort of children, in line with the authors’ hypothesis, high titers of CeD-specific antibodies were associated with severe clinical presentations and histological damage [48]. In a Finnish study, the degree of villous atrophy correlated with signs and symptoms [15]. In another two studies not applying the Oslo Classification, multiple symptoms were associated with high titers of CeD-specific antibodies and severe intestinal damage [14,49]. Two studies involving adult CeD patients failed to observe a significant association between clinical presentations and histological damage [17,50]. On the contrary, in an Iranian study and in a Finnish study, the association between gastrointestinal symptoms and the degree of villous atrophy was proven to be statistically significant [16,51]. In our study, contrary to some of the above-mentioned literature data [14,15,34,48,49], neither intestinal damage nor the titers of CeD-specific antibodies were significantly associated with clinical presentations. The results from the uni- and multivariate analyses corroborated these findings (Table 1 and Table 2).

Our findings call attention to the complexity of factors influencing clinical presentations of CeD: high titers of CeD-specific antibodies and severe villous atrophy do not guarantee clinically severe disease, vice versa, and clinically silent CeD can be associated with a prominent immune reaction.

Although guidelines provide clear instructions on when to suspect CeD, how to carry out diagnostics and how to manage therapy and follow-up, practice does not always match these recommendations. This is particularly true in the developing countries. For example, a recent Indian study showed that the number of cases diagnosed only based on serology was high, while dietary counseling and follow-up is often missing, as well [52]. In Western countries, the recognition of silent or mild, atypical cases has improved a lot, but it is still not optimal [2]. In Hungary, the conditions for adequate diagnostics are given; however, similarly to other middle-European countries, access to dietary counseling and follow-up should be improved [53]. In our academic center, we founded a regional center for CeD care 20 years ago. Our center provides care for the highest number of adult CeD cases in Hungary. We harmonized CeD care, provided continuous access to dietary counseling and set up the Hungarian Coeliac Disease Registry. In addition to providing guideline-adherent care, we aim to call attention to the importance of the diagnostics of non-classical cases (e.g., those with atrophy and cryptogenic/unexplained hypertransaminasemia) and silent cases.

The strengths of our study include its sample size, the accurate case definition and the application of the multivariate analysis, which, to our best knowledge, has not been used to investigate the association between baseline parameters and clinical presentations of CeD. Here, we report the data of a great cohort of Hungarian CeD patients, whereas until now, only descriptive data on small cohorts of Hungarian CeD patients have been published [54]. The limitations of our study include its retrospective nature, the variation in the data recording with time and the limited available data on certain baseline variables (e.g., detailed non-immunological laboratory tests). Furthermore, in this work, we were unable to investigate the potential role of genetics in determining clinical presentations of CeD.

## 5. Conclusions

Our findings, contrary to some of the previous research, confirmed that classical CeD is independently associated with age at diagnosis and calendar year of diagnosis of CeD, whereas sex, intestinal histology and serology at diagnosis were not significantly associated with clinical presentations. This implies that clinicians have to pay even closer attention to the active case-finding strategy applicable in many disciplines in order to recognize atypical, mild CeD cases in the ‘celiac iceberg’. However, further research is warranted to identify factors in the background of age- and calendar year-related trends (e.g., the role of the microbiome, genetics, metabolic factors or changes in the adaptive abilities of the small intestines might be implicated).

## Figures and Tables

**Figure 1 jpm-13-00487-f001:**
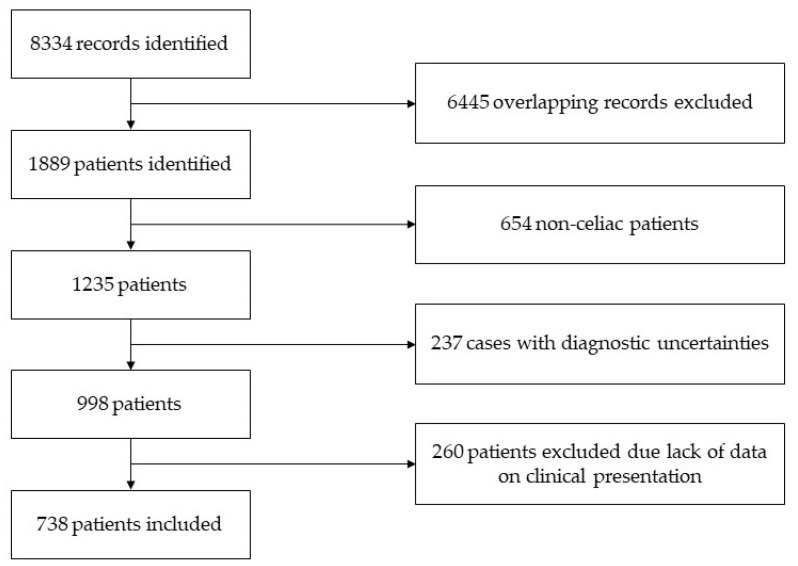
Flow chart of the study.

**Table 1 jpm-13-00487-t001:** Characteristics of patients included.

	Total Cohort of Patients (n = 738)	Patients with Classical Presentation (n = 290)	Patients with Non-Classical Presentation (n = 448)
Age at diagnosis (median and range, years) (continuous)	22.8 ± 17.1	24.4 ± 18.8	21.8 ± 15.9
Age at diagnosis (dichotomous)			
diagnosed in childhood	362	122	240
diagnosed in adulthood	376	168	208
Sex			
males	194	67	127
females	544	223	321
tTG IgA at diagnosis			
negative	39	17	22
low positive (1.0–9.9 times of the upper normal level)	159	53	103
high positive (at least 10 times of the upper normal level)	407	139	268
tTG IgG at diagnosis			
negative	189	64	125
low positive (1–9.9× titer)	287	102	185
high positive (at least 10× titer)	107	33	74
EMA (IgA) at diagnosis			
negative	48	21	27
weak positive	16	4	12
strong positive	523	170	353
EMA (IgG) at diagnosis			
negative	112	37	75
weak positive	26	10	16
strong positive	274	88	186
Histology at diagnosis			
Marsh 1	6	2	4
Marsh 2	11	4	7
Marsh 3a	61	22	39
Marsh 3b	126	50	76
Marsh 3c	258	103	155
Marsh 3 (not classified otherwise)	14	6	8
not classified per Marsh/Marsh-Oberhuber criteria	50	23	27
non-interpretable or non-specific changes	17	8	9
Calendar period			
≤1998	56	43	13
1999–2007	176	78	98
2008–2019	506	169	347

tTG: anti-tissue transglutaminase antibody, EMA: anti-endomysial antibody.

**Table 2 jpm-13-00487-t002:** Associations between baseline parameters and clinical presentation of celiac disease (multivariate logistic regression analysis).

	Total Number of Patients	Number of Patients with Classical Presentation	Odds Ratio	95% Confidence Interval	*p*-Value
Age at diagnosis (years)	392	135	1.02	1.01–1.04	<0.001
Sex	392	135			
Male	91	25	1.00 (ref.)		
Female	301	110	1.34	0.79–2.32	0.290
Calendar year of diagnosis (years)	392	135	0.93	0.89–0.98	0.006
Histology	392	135			
Marsh 1+2	15	5	1.00 (ref.)		
Marsh 3a+b	151	50	0.90	0.90–3.12	0.857
Marsh 3c	226	80	1.20	0.39–4.11	0.758
tTG IgA	392	135			
negative	24	12	1.00 (ref.)		
low positive	102	40	0.63	0.25–1.60	
high positive	266	83	0.52	0.22–1.28	

tTG: anti-tissue transglutaminase antibody.

## Data Availability

Data are contained within the article or Appendix A.

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
