# Peer review of "Clinical Presentation Is Dependent on Age and Calendar Year of Diagnosis in Celiac Disease: A Hungarian Cross-Sectional Study"

_jpm, 2023, doi:10.3390/jpm13030487_

Round 1

Reviewer 1 Report

Review for the manuscript Integrative Clinical Presentation is Dependent on Age and Calendar Year 2 of Diagnosis in Celiac Disease: A Hungarian Cross-Sectional 3 Study submitted to JPM.

Dear Editor, thank you for the opportunity to review this manuscript. After careful evaluation, I suggest some modifications before it can be published.

Overall comments: This is an interesting and well-conducted study that had as its main Objective to investigate the association of clinical presentation of celiac disease in a retrospective, cross-sectional study.

ABSTRACT

            The abstract is fine. However, I suggest a sentence before the first one giving an Introduction sense.

KEYWORDS

            I suggest including at least two more keywords.

INTRODUCTION

            This section is rich with relevant information. However, I miss the inclusion of newer references. I suggest including references published in 2022 and 2023. There are so many good studies that the authors can find in PUBMED or Cochrane. As examples, the authors can check very good studies in PUBMED, such as:

1)    Japanese Pediatricians and Celiac Disease. Kumagai H.Pediatr Int. 2023 Feb 17:e15509. doi: 10.1111/ped.15509.

2)    Taking a "Second Look" at the Incidence of Pediatric Celiac Disease. Egberg MD.Am J Gastroenterol. 2023 Feb 15. doi: 10.14309/ajg.0000000000002136.

3)    Patterns of practice in the diagnosis, dietary counselling and follow-up of patients with celiac disease- A patient-based survey. Mehtab W, Agarwal H, Ghosh T, Chauhan A, Ahmed A, Singh A, Vij N, Singh N, Malhotra A, Ahuja V, Makharia GK.Indian J Gastroenterol. 2023 Feb 13. doi: 10.1007/s12664-022-01296-7.

4)    Celiac Disease Genetics, Pathogenesis, and Standard Therapy for Japanese Patients. Tamai T, Ihara K.Int J Mol Sci. 2023 Jan 20;24(3):2075. doi: 10.3390/ijms24032075.

Moreover, I would suggest that the authors could go beyond the “CeD affects approximately 1% of the population worldwide” as we see in lines 38-39. Is this percentage the same for Europe, Asia and Americas?

In this section, please, also include the influence of Ced in the quality of life of the patients.

METHODS

            This section was adequately described. However, the informationwe find at the end of the manuscript “The study was conducted in accordance with the Declara- 214 tion of Helsinki and approved by the Regional and Local Research Ethics Committee of University 215 of Pécs, Pécs, Hungary (Ref No 6918)” should be seen in the Methods section.

            I suggest that the authors expand the description of the inclusion/exclusion criteria in the study. Were other associated diseases the reason for exclusion? There were patients above the age predicted in Table 1 (22.8±17.1, for example). It seems to me that those included are relatively young patients.

            Was there a sample calculation for the sample? Please, justify.

RESULTS

DISCUSSION

            This section is adequate, but I have some comments:

1)    In lines 144-145 we can see “Our study aimed to investigate if baseline parameters are associated with clinical presentation in CeD”. This sentence belongs to the Objective of the study and is repeated here. I suggest removing it.

2)    In the lines 165-166 we can read “Regarding the association with sex, the more severe classical clinical presentation in women was not confirmed by our study [10].” Please, include more references that show the more severe clinical presentation is seen in women.

3)    In line 181-183 we find “In our study, contrary to some literature data, neither intestinal damage, nor the titres of CeD-specific antibodies were significantly associated with clinical presentation. Results from uni- and multivariate analyses corroborated”. Please, include references for the sentence “contrary to some literature data”.

4)    When the authors say that “Besides with in this work, we were unable to investigate the potential 196 role of genetics in determining clinical presentation of CeD (21,41)”. Why citing these references. I see that these references are Bajor, J., et al. (2019) and Bajor, J., et al (2019). However, I think that the limitations belong to this present study, and it is not necessary to refer to previous publications of the authors (despite being related to this study).

Furthermore in this sentence I suggest replace “Besides with in this work…” for “Besides within this work…”

5)    How could the results of this study contribute to the patients and clinicians?

CONCLUSION

            This section is adequate.

REFERENCES

            As pointed out above, I suggest including newer references in the Introduction and in the Discussion section.

Reviewer 2 Report

In this retrospective, cross-sectional study, the authors aimed to explore the associates of clinical presentation of celiac disease (CD). They assessed associations of baseline variables with clinical presentation. Overall, 738 CD patients were studied. In univariate analysis, patients with classical CD were more common in the latest calendar period (p<0.001) and tended to be older (p=0.056), but they did not observe a significant association between the clinical presentation and sex, serology, or histology at diagnosis. In multivariate analysis, only age at diagnosis and calendar year were independently associated with clinical presentation. They concluded that classical CD is independently associated with age at diagnosis and calendar year of diagnosis of CeD, whereas sex, intestinal histology and serology at diagnosis were not significantly associated with clinical presentation.

The study is of interest since clinical presentations of CD display different features worldwide and are of clinical relevance in describing regional differences. However, since the diagnostic process has changed in the last decades and we are now more confident with atypical presentations as well as with extraintestinal disease manifestations, in my opinion, adjunctive pieces of information are needed as well as recalling additional clinical features of non-classical celiac disease.

-Materials and Methods: Regarding the diagnosis of CD, the authors should specify whether they also tested total IgA serum levels to exclude IgA deficiency as well as histological assessment of multiple and rightly oriented duodenal biopsies as recommended by all current guidelines (Current guidelines for the management of celiac disease: A systematic review with comparative analysis. World J Gastroenterol. 2022 Jan 7;28(1):154-175. ).

-Introduction: regarding the very important issue of clinical presentation of celiac disease, a topic of major clinical relevance worth mentioning is related to the associated autoimmune diseases and corresponding autoantibodies that have been extensively described in celiac disease patients, such as antineuronal and antiganglioside antibodies in patients with neurological disorders, as previously showed (Sera of patients with celiac disease and neurologic disorders evoke a mitochondrial-dependent apoptosis in vitro. Gastroenterology. 2007;133(1):195-206; Anti-ganglioside antibodies and celiac disease. Allergy Asthma Clin Immunol. 2021 May 28;17:53.), as well as other highly CD-specific autoantibodies of prognostic significance such as anti-actin IgA antibodies which are significantly associated to severe mucosal damage (villous atrophy) as previously demonstrated (Anti-actin IgA antibodies in severe coeliac disease. Clin Exp Immunol. 2004 Aug;137(2):386-92.). 

-Conclusions: the authors stated that "These implicate that clinicians have to pay even closer attention to the active case finding strategy applicable in many disciplines in order to recognize atypical, mild CeD cases in the ‘celiac iceberg’. This is a very clinically relevant point of the study and, taking into account other well-described patients with a higher risk of celiac disease, also silent patients with atopy or patients with cryptogenic/unexplained hypertrasaminasemia should be tested for celiac disease as previously described (Prevalence of silent coeliac disease in atopics. Dig Liver Dis. 2000;32(9):775-9; Anti tissue transglutaminase antibodies as predictors of silent coeliac disease in patients with hypertransaminasaemia of unknown origin. Dig Liver Dis. 2001;33:420-5.).

Round 2

Reviewer 1 Report

Dear authors,
Thank you for performing the corrections to the manuscript, which is now ready for publication.

With regards

Reviewer 2 Report

The author properly addressed the raised points.